# Combination of High-Pressure Processing and Freeze-Drying as the Most Effective Techniques in Maintaining Biological Values and Microbiological Safety of Donor Milk

**DOI:** 10.3390/ijerph18042147

**Published:** 2021-02-22

**Authors:** Sylwia Jarzynka, Kamila Strom, Olga Barbarska, Emilia Pawlikowska, Anna Minkiewicz-Zochniak, Elzbieta Rosiak, Gabriela Oledzka, Aleksandra Wesolowska

**Affiliations:** 1Department of Medical Biology, Faculty of Health Sciences, Medical University of Warsaw, 14/16 Litewska St., 00-575 Warsaw, Poland; sylwia.jarzynka@wum.edu.pl (S.J.); kamila.strom@wum.edu.pl (K.S.); o.barbarska@gmail.com (O.B.); anna.minkiewicz@wum.edu.pl (A.M.-Z.); gabriela.oledzka@wum.edu.pl (G.O.); 2Institute of High Pressure Physics of the Polish Academy of Sciences, ul. Sokolowska 29/37, 01-142 Warsaw, Poland; emilia.pawlikowska@gmail.com; 3Institute of Human Nutrition Sciences, Warsaw University of Life Sciences-SGGW, Nowoursynowska 159c St., 02-776 Warsaw, Poland; elzbieta_rosiak@sggw.pl; 4Laboratory of Human Milk and Lactation Research at Regional Human Milk Bank in Holy Family Hospital, Department of Medical Biology, Faculty of Health Sciences, Medical University of Warsaw, 14/16 Litewska St., 00-575 Warsaw, Poland

**Keywords:** safety storage of donor milk, high-pressure processing, freeze-drying

## Abstract

Background: Human milk banks have a pivotal role in provide optimal food for those infants who are not fully breastfeed, by allowing human milk from donors to be collected, processed and appropriately distributed. Donor human milk (DHM) is usually preserved by Holder pasteurization, considered to be the gold standard to ensure the microbiology safety and nutritional value of milk. However, as stated by the European Milk Banking Association (EMBA) there is a need to implement the improvement of the operating procedure of human milk banks including preserving and storing techniques. Aim: The purpose of this study was to assess the effectiveness and safety of the selected new combination of methods for preserving donor human milk in comparison with thermal treatment (Holder pasteurization). Methods: We assessed (1) the concentration of bioactive components (insulin, adiponectin, leptin, activity of pancreatic lipase, and hepatocyte growth factor) and (2) microbiological safety in raw and pasteurized, high-pressure processed and lyophilization human breast milk. Results: The combination of two techniques, high-pressure processing and freeze-drying, showed the best potential for preserving the nutritional value of human milk and were evaluated for microbiological safety. Microbiological safety assessment excluded the possibility of using freeze-drying alone for human milk sample preservation. However, it can be used as a method for long-term storage of milk samples, which have previously been preserved via other processes. Conclusion: The results show that high-pressure treatment is the best method for preservation that ensures microbiological safety and biological activity but subsequent freeze-drying allowed long-term storage without loss of properties.

## 1. Introduction

Human milk, which contains a unique composition of essential nutrients, antibodies, growth factors, hormones, oligosaccharides, and favourable microbiota, provides optimal nutrition for newborns [1]. The nutritional composition of human milk has been shown to ensure optimal development of infants, especially preterm infants; hence, the World Health Organization and the American Academy of Pediatrics recommend feeding babies with the mothers’ milk, particularly during the first 6 months of life [2,3]. Obtaining human milk from donors and storing it in milk banks has become an increasingly common practice. This source of top-quality human milk offers preterm babies the chance to have access to optimal nutrition rich in bioactive components such as hormones, enzymes, and growth factors in the first weeks of their lives [4]. Mother’s milk is an easily absorbable food that lowers the risk for conditions such as necrotizing enterocolitis and subsequent sepsis [5].

Optimal utilization of the excess donor human milk (DHM) requires adequate processing methods and storage conditions. The most commonly used method, recommended by the European Milk Bank Association (EMBA), for human milk banks (HMB) treatment is low-temperature (62.5 °C) long-duration (30 min) pasteurization, also known as Holder pasteurization (HoP) [6,7]. This treatment method ensures the microbiological safety of human milk, as it inactivates bacteria, fungi, and some viruses. However, *Bacillus* spp. spores have been shown to be resistant to high temperatures [8,9,10]. One drawback of HoP is the fact that it compromises the structure of human milk nutrients, such as vitamins, cytokines, immunoglobulins, lysozyme, lactoferrin, lipoprotein lipase, and hormones (e.g., insulin) [11,12]. Human milk treated with HoP must be stored below zero degrees Celsius, which has an additional negative impact on its biological and therapeutic values [13]. Thermal HoP has two main problems: first, the degradation of valuable ingredients like antibodies, enzymes or vitamins and second, non-selective reduction of both potentially pathogenic microorganisms and probiotic microbiota. However, the main goal for milk banks is the long-term storage of human milk samples, without loss of biological activities and with preservation of microbiological purity [9,10].

The fact that HoP lowers the therapeutic value of human milk prompts a search for new alternative treatment methods that will ensure the microbiological safety of donor milk, while preserving its bioactive components. High-pressure processing (HPP), or pascalization, is an efficient method of food preservation [13,14,15]. Human milk exposed to high pressure (100–1000 MPa) exhibits better nutritional and bioactive properties than that exposed to high temperatures [16]. A review of the relevant literature indicates that HPP may be a promising alternative to HoP as it reduces the loss of immune factors, cytokines, and enzymes contained in breast milk. Wesolowska has also demonstrated that unlike thermal processing, HPP preserves the amount of IgA and lysozyme in human milk [17,18]. Furthermore, it has been shown that the use of HPP (400 MPa) effectively inactivates the evaluated pathogens [19].

A plausible alternative to low-temperature storage of microbiologically pure human milk rich in bioactive components may be freeze-drying, or lyophilization. This treatment method helps extend the shelf-life of foods by diminishing their water content to 1–3% at freezing temperatures. According to literature, freeze-drying significantly reduces the number of mesophiles and spore-forming *Bacillus* species in human milk, whereas it has no significant effect on the number of *Staphylococcus aureus* or *Enterococci* [20,21]. Freeze-drying of human milk causes no degradation of most of the nutrients, vitamins, or proteins, and the taste and flavor remain unchanged following rehydration [21,22,23]. Apart from the findings by Salcedo et al. (2015) on the successful reduction of germs in breast-milk after freeze-drying, it was shown that storage of freeze-dried milk did not show a significant influence upon several biological properties of human milk such as total antioxidant capacity and fatty acid profile of milk [21,22,23,24]. The storage and transport of milk powder is also considerably less complicated in the hospital setting and makes possible longer storage than in the case of pasteurized milk, which must be frozen [21,22,25].

Furthermore, the powder can be easy to reconstitute at a given volume to obtain the required caloric and nutrition value of milk for preterm infants [25,26].

Given those advantages, we decided to combine those methods to optimize the human milk treatment that would ensure prolonged microbiological safety and preserve bioactive components of donor milk to the greatest extent comparable with that offered by HoP.

Our study analyzed the selected bioactive components and microbiological safety of stored human milk samples subjected to standard pasteurization (HoP); a new, modified process of pascalization (HPP); freeze-drying; and a combination of the latter two, promising in human milk preservation, which may extend the time of using a milk sample stored in human milk banks. Innovative HPP parameters that have been described by Wesolowska were used [18]. We also evaluated whether milk samples that retained their bioactive values after being subjected to treatment processes were also microbiologically safe, and if so, for how long? We chose lipase, leptin, adiponectin and insulin as the most important factors in energy metabolisms, food intake and appetite regulation, which constitute a leading role in newborns’ nutrition. Lipase and adiponectin additionally stimulate the immune system of neonates and have also cardioprotective effects and an intake role in bone formation. Hepatocyte growth factor (HGF) is involved in regulating the growth of intestinal cells in the newborn. Microbiological tests were performed for microorganisms that could cause gastrointestinal, respiratory tract, and systemic infections, including meningitis and sepsis, in neonates and infants. Due to the increase in contamination of milk with potentially pathogenic bacteria, in our microbiological tests, we used the method of milk fortification with the selected pathogen and then treating these samples with various techniques that can be used in the safe preparation and storage of human milk in milk banks. The goal was to find the safest method in terms of preserving important hormones and metabolic enzymes, and the most effective in terms of microbiological purity.

## 2. Materials and Methods

### 2.1. Experimental Design and Sample Preparation

The evaluated milk samples, originally collected from donors between the second and sixth week of lactation, were obtained from the Regional Human Milk Bank in Warsaw at the Holy Family Hospital. Donors had been given standard instructions about the best practices for expressing and collecting milk. Obtained milk samples were pooled separately for the two experiments (microbiological safety assessment and determination of bioactive components) (Figure 1). A total volume of 7700 mL of thawed, pooled donated human milk was used for microbiological tests. A total volume of 2100 mL of fresh, non-frozen, pooled donated milk was used for the determination of bioactive components. Frozen milk samples were stored at −20 °C until processing. Fresh milk samples were refrigerated at 4 °C and delivered to the human milk bank within 24 h under refrigerated conditions.

### 2.2. Determination of Bioactive Components

Six independent experiments were carried out using fresh unfrozen milk samples collected from 3–4 donors pooled for each one. Each milk pool was divided, and the resulting samples (apart from a sample of raw, untreated milk) were subjected to the below techniques of milk preservation evaluated in this study:(a)raw, untreated milk (Raw),(b)holder pasteurization (HoP),(c)high-pressure processing (HPP) (450 MPa, 21 °C, 15 min),(d)freeze-drying (Lyo) (Raw + Lyo),(e)holder pasteurization followed by freeze-drying (HoP + Lyo),(f)high-pressure processing followed by freeze-drying (HPP + Lyo).

The levels of the selected bioactive components were analyzed in each sample. Bioactive component levels were measured 10 h after milk processing (time 0) and then, again, after the samples had been stored for 3 months. The shelf life of treated milk was determined based on the stability of bioactive compounds in pasteurized milk evaluated within 10 h (time 0) after processing. The experimental design is presented in Figure 1. All the evaluated bioactive components of milk were analyzed by enzyme-linked immunosorbent assay (ELISA). The assays were performed in duplicate. The concentrations of these components were analyzed with commercial ELISA kits for Human Leptin Quantikine (Bio-techne, Ltd., Abingdon, UK); Human HMW Adiponectin/Acrp30 Quantikine (R&D Systems, Inc., Minneapolis, MN, USA); Human HGF Quantikine (R&D Systems, Inc., Minneapolis, MN, USA); Insulin (DRG Instruments GmbH, Marburg, Germany); and QuantiChrom Lipase Assay Kit (Bioassay Systems, Hayward, CA, USA). Optical density was measured with a microtiter plate reader (Synergy HTX multimode reader, Biotek^®^, Winooski, VT, USA). The Gen5 Data Analysis Software (Biotek^®^, Winooski, VT, USA) was used for data analysis.

### 2.3. Holder Pasteurization

Holder pasteurization (HoP) was performed at the Regional Human Milk Bank in Warsaw at the Holy Family Hospital on an automatic Human Milk Pasteurizer S90 Eco (Sterifeed, Medicare Colgate Ltd., Post Cross Business Park, Kentisbeare, Cullompton, Devon, UK). Fifty-milliliter milk samples were treated for 30 min at 62.5 °C according to the standard operating pasteurization protocol.

### 2.4. High-Pressure Processing

High-pressure processing (HPP) was performed at the Institute of High Pressure Physics, Polish Academy of Sciences, using a U 4000/65 apparatus (designed and produced by Unipress Equipment, Warsaw, Poland) as per methods described previously [18]. Samples were treated with a high pressure of 450 MPa for 15 min at 21 °C.

### 2.5. Freeze-Drying

The process of freeze-drying or lyophilization of human milk was performed at WPPH Elena (Kokanin, Poland) on a vacuum method. The lyophilization process, which involves freezing the water content and, subsequently, causing the ice to sublime, was carried out by freezing fresh raw human milk to −20 °C and then freeze-drying it at 30–40 °C for 24–72 h under vacuum conditions. As a result, a dry product was obtained in powdered form.

### 2.6. Microbiological Safety Assessment

Major pathogens, such as *Staphylococcus aureus, Listeria monocytogenes*, and *Cronobacter sakazakii*, were selected according to the European Milk Bank Association (EMBA) recommendations to assess the safety of alternative methods of donor milk preservation [6]. Moreover, all the samples were tested for the Gram-negative intestinal *Enterobacteriaceae* family, in particular, *Escherichia coli* and vegetative *Bacillus cereus* rods. Raw milk (100 mL) was also tested for the native, or background microbiota (aerobic and mesophilic bacteria) and major pathogens. Microbiological analysis of raw milk was conducted at 4 °C within 30 min from pooling.

Frozen raw donor milk was thawed, pooled, and subjected to holder pasteurization (30 min, 62.5 °C) to eliminate contamination and background microflora. The purity of milk after processing was detected via microbiological testing on a 100 mL sample. The resulting, pasteurized milk was divided into equal samples. Several milk samples were then inoculated with each of the following selected reference bacterial strains: *Escherichia coli* (ATCC 25922), *Staphylococcus aureus* (ATCC 33862), *Listeria monocytogenes* (ATCC 7644), *Cronobacter sakazakii* (ATCC 51329), and *Bacillus cereus* (ATCC 14579), to a final concentration of 10^6^ colony-forming units (CFU) in 1 mL of human milk. According to EMBA guidelines, the fortification method is simple and the best technique in vitro to assess the possibility of using various methods of milk decontamination in the conditions of milk banks [6]. The samples inoculated with different reference bacterial strains were incubated according to the established standards and then subjected to each of the evaluated techniques of human milk preservation (listed in points b–f mentioned above). All samples were processed in five replicates, stored for 3 months and 6 months frozen at −20 °C or in freeze-drying form at refrigerator temperature. (Figure 1). Microbiological tests were carried out following the European standards for testing food products on thawed samples or on hydrated samples after freeze-drying before processing., in terms of the total number of aerobic mesophilic bacteria (Plate Agar Count, Merck, Darmstadt, Germany, EN ISO 4833-2:2013); Gram-negative bacteria, such as the *Enterobacteriaceae* family, including *Escherichia coli* (Crystal-Violet Neutral Red Bile Glucose Agar, BioMaxima, Lublin, Poland, EN ISO 21528-2:2017); coagulase-positive staphylococci, such as *Staphylococcus aureus* (Baird Parker Agar, BioMaxima, Lublin, Poland, EN ISO 6888-1:2004); Gram-positive rods, such as *Listeria monocytogenes* (Fraser Medium, BioMaxima, Lublin, Poland, EN ISO 11290-2:2017-2); Gram-negative rods, such as *Cronobacter sakazakii* (ESIA, BioMaxima, Lublin, Poland, EN ISO/TS 22964:2017); and sporulating bacteria, such as *Bacillus cereus* (MYP Agar, BioMaxima, Lublin, Poland, EN ISO 7932:2004).

### 2.7. Statistical Analysis

All statistical analyses (including the generation of the relevant diagrams) were performed using GraphPad Prism version 6.00 for Windows (GraphPad Software, La Jolla, CA, USA, www.graphpad.com (accessed on 20 February 2020). The differences between the results of all evaluations were analyzed by *t*-test and one-way analysis of variance (ANOVA) with post hoc Tukey’s test. Statistical significance was assumed at a *p*-value of <0.05.

## 3. Results

### 3.1. Determination of Bioactive Components in Treatment Milk Samples

The concentrations of the bioactive components are presented as percentages with respect to raw milk, which represents 100% (Figure 2). The most marked decrease in the concentration of bio-components especially hepatocyte growth factor (HGF), lipase, and leptin, was observed after HoP and HoP + Lyo treatments. These results were statistically significant. Following holder pasteurization, the concentration of HGF, lipase, and leptin in the processed-milk was decreased by 97%, 96%, and 66%, respectively, compared with raw milk. Following HoP + Lyo, the milk concentrations of HGF, lipase, and leptin were decreased by 97%, 95% and 72%, respectively. For insulin and adiponectin, the decrease in concentrations was between 10% and 24%. The concentration of the bioactive components in milk following HPP decreased by only 21% for HGF and by 21% for lipase and increased by 1% for leptin and by 10% for insulin. The only bioactive compound on which HPP had a highly unfavorable effect was adiponectin (85% decrease). Human milk samples processed by a combination of HPP and freeze-drying (HPP + Lyo) showed a decrease in HGF, insulin, and lipase concentrations by 28%, 7%, and 24%, respectively. Leptin increased by about 16%, whereas adiponectin decreased by 86%. Freeze drying (Lyo) showed the most favorable results in terms of the concentrations of the bioactive compound.

Evaluation of the concentration of the bioactive components in milk samples subjected to a combination of pascalization and freeze-drying (HPP + Lyo) after storage for 3 months is presented in Figure 3. Adiponectin, HGF, insulin, and lipase levels were decreased following storage. The only bioactive components whose levels changed significantly in milk samples subjected to HPP + Lyo were adiponectin (at time 0, *p* < 0.0001, and month 3, *p* < 0.0001) and HGF (at month 3; *p* = 0.0049). An unfavorable effect of long storage on milk samples was observed only in terms of adiponectin levels. The changes in insulin and lipase levels were not statistically significant at either of the two storage time points. In the case of leptin, we observed an increase in its concentration in HPP + Lyo milk samples after processing and after 3 months of storage, but the results were not statistically significant and require further confirmation. The combination of pascalization and lyophilization seems unlikely to considerably affect lipase, leptin, insulin, or HGF levels in stored milk samples. However, this study showed an upward trend in leptin concentration and a downward trend in adiponectin concentration in stored milk samples.

### 3.2. Microbiological Safety

#### 3.2.1. Native Microbiota

The total number of native microbiota in raw human milk was approximately 6.9 × 10^4^ CFU/mL, including *Staphylococcus aureus* (2.8 × 10^4^ CFU/mL), *Enterobacteriaceae* family, such as the genus *Enterobacter* (1.8 × 10^4^ CFU/mL)*,* and *Bacillus cereus* (6.5 × 10^3^ CFU/mL). No growth of *Escherichia coli*, *Listeria monocytogenes*, or *Cronobacter sakazakii* was observed. According to EMBA recommendations, human milk samples should contain no more than 10^5^ CFU of non-pathogenic microorganisms per milliliter before pasteurization [7]. The results from Holder pasteurization indicate that the vegetative forms of native microbiota are destroyed by this treatment. Our study confirmed this, as milk samples were found to be free of vegetative germs after pasteurization.

#### 3.2.2. Microbiological Purity after High-Pressure Processing and Freeze-Drying in Storage Milk Samples

Human milk samples free from microorganisms were subjected to inoculation with bacterial strains at an initial concentration of 10^6^ CFU/mL. After inoculating milk samples, we confirmed the actual bacterial concentration at the following levels: *Escherichia coli* 1.2 × 10^7^ CFU/mL, *Staphylococcus aureus* 6.9 × 10^6^ CFU/mL, *Listeria monocytogenes* 8.2 × 10^7^ CFU/mL, *Cronobacter sakazakii* 8 × 10^5^ CFU/mL, and *Bacillus cereus* 4.0 × 10^5^ CFU/mL. The growth of the selected microorganisms in freeze-dried milk samples was observed at 3 time points (at time 0, at 3 months of storage, and at 6 months of storage) (Table 1).

Compared with the initial concentration, we observed a decrease in *Escherichia coli* growth by almost 3 orders of magnitude (initial concentration of 1.2 × 10^7^/log_10_ 7.07 vs. concentration of 2.1 × 10^4^/log_10_ 4.33 following Lyo at time 0; *p* < 0.05). This effect decreased as the storage time increased (*p* < 0.05) (Figure 4). There was also a decrease in *Bacillus cereus* growth by 3 orders of magnitude at 3 months of storage after Lyo (from 4 × 10^5^ at time 0 to 1.2 × 10^2^ CFU/mL at 3 months of storage), that was statistically significant (*p* = 0.042). After 6 months of storage, the freeze-dried human milk samples inoculated *Bacillus cereus* showed a complete reduction (by 100%) in the growth of vegetative forms of this species (Figure 4). The concentration of *Listeria monocytogenes* decreased, but only slightly (from 8.2 × 10^7^ to 1.8–2.6 × 10^7^ CFU/mL, *p* < 0.0001), following lyophilization of milk samples inoculated with this strain. This phenomenon was demonstrated in all lyophilized milk samples stored for 3 months. A similarly inadequate effect of lyophilization was observed by the lack of growth inhibition in both *Staphylococcus aureus* and *Cronobacter sakazakii* strains (Table 1).

HPP was the best method for inactivating vegetative forms of the pathogens present in the inoculated milk samples. Moreover, samples subjected to HPP and those subjected to HPP + Lyo were free from pathogenic bacteria even after 6 months of storage. Both milk treatment methods (HPP and HPP + Lyo) yielded complete eradication of bacterial growth at all the evaluated storage time points (Table 1). The inactivation efficiency (IE) (the decrease in bacterial growth by orders of magnitude following HPP and HPP + Lyo compared with initial bacterial concentration) was: 7.07 log for *Escherichia coli*, 6.83 log for *Staphylococcus aureus*, 7.91 log for *Listeria monocytogenes*, 5.9 log for *Cronobacter sakazakii*, and 5.3 log for *Bacillus cereus* (Table 1).

## 4. Discussion

According to EMBA recommendations based on multi-centre consensus, donated human milk should contain no more than 10^3^–10^4^ CFU of bacteria per mL [11,27]. In our study, raw human milk sample cultures yielded native microbiota (background microbiota) count of 4.84 log_10_ CFU/mL; this included pathogenic faecal *Enterobacteriaceae* family and *Bacillus cereus* species. Global reports indicate human milk sample contamination with the genus *Bacillus* as an important safety concern associated with human milk donation [9,10]. Nonetheless, due to the spore-forming ability of these bacteria, reliable diagnostic investigations are difficult. Studies from around the world also indicate the risk of human milk contamination with *Cronobacter sakazakii* [20,28]. According to microbiological examinations of human milk stored in HMB, there was a marked increase in *Bacillus cereus* contamination in human milk. Adjidé et al., analyzed 1585 batches of donated human milk and found that about 27.3% of the samples had non-compliance with the microbiological quality. *Bacillus cereus* was the main cause of non-compliance [29].

Therefore, the global target of milk banks is the application of the best technology for microbiological safety and diminution of biological ingredient losses. The aim of this study was to determine the optimal method for preserving and storing donor human milk at banks. Feeding infants, especially preterm infants, with human milk is an important therapeutic practice for them. Human milk preservation methods commonly used in milk banks are based mainly on HoP. Our findings suggest that HoP leads to a complete (100%) eradication of milk vegetative microbiota (including the valuable probiotic bacteria) and lowers the levels of bioactive components (including HGF and lipase) by up to 97%. Other authors have reported similar findings [30,31]. HoP has been shown to result in a complete eradication or decrease in the counts of endogenous bacteria, namely *Escherichia coli* and *Staphylococcus aureus* [32]. Our in-house studies demonstrated that human milk samples intended for inoculation constituted a “sterile” medium for the evaluated, potentially pathogenic, bacteria. The most beneficial method for maintaining the bioactive components of human milk is lyophilization alone. According to the results reported by Oliviera, the nutritional value after 3 and 6 months of storage remains within the acceptable standards; however, there is only a slight reduction in potassium and copper [26]. Moreover, the process of lyophilization fails to ensure microbiological purity, which disqualifies freeze-drying alone for HMB. Additionally, the lyophilization of sterile food samples prevents bacterial contamination of these products in long-term storage. It is a method that can only be used for storage of a finished product, like human milk fortifiers for feeding very low birth weight (VLBW) infants. Oliviera’s current research confirms the possibility of formulation and utilization of a concentrate from lyophilized human milk [26]. It is a matter of preparing a sterile biological product that can be lyophilized. A proposal to solve this problem has been presented in this study. In this study, human milk samples inoculated with bacteria and later freeze-dried and stored for a period from 3 to 6 months were shown to yield bacterial growth at the level of the initial bacterial concentration. Such results were seen with human milk samples inoculated with *Staphylococcus aureus*, *Listeria monocytogenes*, and *Cronobacter sakazakii*. In the case of *Escherichia coli*, lyophilized milk samples showed a decrease of three orders of magnitude in microbial concentration in comparison with the initial bacterial concentration (Figure 4). The *Bacillus cereus* bacteria were absent in lyophilized milk samples after storage for 6 months, suggesting a complete eradication of these bacterial strains. This is an interesting finding that presents an opportunity for further research. Confirmation of these results requires additional studies. Some authors indicated the possibility of the inhibition of growth of *Bacillus cereus* as well as other Gram-positive bacteria after freeze-drying. These studies indicated only a reasonable growth inhibition in *Bacillus cereus* species following lyophilization. This topic needs further research. Our findings are consistent with those of other authors who used bacterial culture lyophilization to maintain strain viability during long-term storage [33]. The results of this study on human milk lyophilization demonstrated the possibility of utilizing this preservation method for storing human milk, without significantly altering its content in bioactive components.

The search for novel, effective methods of human milk preservation that maintain its microbiological purity, hence its safety, led us to the use of a combination of HPP (pascalization) and lyophilization. HPP alone is a promising non-thermal method for pathogen inactivation. As part of our in-house studies, this milk-processing method led to the complete eradication of vegetative *Bacillus cereus* in all the evaluated human milk samples in comparison with the inoculum. The limitation of this project is that we did not investigate the presence of *Bacillus cereus* spores and the impact of HoP and HPP on the spores. These results confirmed the higher efficacy of HPP in controlling pathogenic microbiota in comparison with HoP. Additionally, earlier studies conducted by our group have evaluated certain parameters of HPP to preserve the therapeutic value of human milk [17,18]. The HPP modification used in our in-house studies included a unique combination of parameters, i.e., a 15-min exposure to 450 MPa at 21 °C [34]. In the current study we conducted the first analyze of human milk to be in line with the methodology of EMBA in terms of the evaluation of new technologies. In particularly we applied fortifications using preferred bacterial strains and their concentration in human milk. To demonstrate a statistically significant reduction in bacterial counts, i.e., a high IE, before and after pascalization. The microbial count reduction was observed in the following strains: *Escherichia coli* (by log 7.07 CFU/mL), *Staphylococcus aureus* (by log 6.83 CFU/mL), *Listeria monocytogenes* (by log 7.91 CFU/mL), *Cronobacter sakazakii* (by log 5.9 CFU/mL), and *Bacillus cereus* (by log 5.3 CFU/mL (Table 1).

Viazis et al. reported similar results regarding the antibacterial effect of HPP that was achieved with 400 MPa [35]. Vegetative forms of *Listeria monocytogenes* and *Streptococcus agalactiae* were inactivated after 2–4 min of HPP. After 30 min, HPP led to eradication of *Escherichia coli* and *Staphylococcus aureus*. Other authors have reported *Enterobacter* spp. eradication after 5 min of HPP [36]. Moreover, they demonstrated comparable inactivation efficiency at various pressures (400, 500, and 600 MPa). The most recent report by Demazeau et al. also demonstrated a complete eradication of *Bacillus cereus* and *Staphylococcus aureus* following HPP at a relatively low pressure of 350 MPa, although the temperature was considerably higher at 38 °C and the duration was considerably longer at 25 min than that reported in our studies (21 °C, 15 min) [19]. The HPP modifications that were introduced through our in-house studies may have resulted in one of the most effective methods of human milk preservation. Moreover, treatment of human milk via HPP led to only slight changes in its bioactive components (1%, 10%, and 21% reduction in leptin, insulin, and lipase levels, respectively) (Figure 2). The only unfavorable change was a considerable reduction in adiponectin levels (by 85%). However, apart from microbiologically safe milk samples rich in bioactive components, human milk banks are particularly interested in terms of the possibility of long-term storage. Therefore, our study evaluated long-term storage of human milk previously subjected to the modified HPP (450 MPa, 21 °C, 15 min).

However, the most important results are related to these milk samples which were treated with HPP and then lyophilized (HPP + Lyo). We observed complete microbiological purity of the evaluated HPP + Lyo milk samples after 3 and 6 months of storage; complete reduction (inactivation efficiency of 100%) in the growth of *Staphylococcus aureus*, *Escherichia coli*, *Listeria monocytogenes*, *Cronobacter sakazakii*, and *Bacillus cereus* was observed (Table 1). Meanwhile, the content of the evaluated bioactive components was satisfactory, i.e., the decrease in the levels of individual components did not exceed 30% (leptin 16%, HGF 28%, insulin 7%, and lipase 24%) at 3 months of storage. The only human milk component that showed a significant decrease was adiponectin, which indicated the possibility of a genuine effect of HPP + Lyo on this component of human milk (Figure 3). Other authors have also reported good results in terms of the nutritional content of lyophilized human milk [24,26]. Oliveira et al. demonstrated preservation of osmolality and stability of nutritional content in freeze-dried human milk samples that were subsequently stored for 3 and 6 months [26]. Martysiak-Zurowska et al., also reported that freeze-drying does not have a significant negative impact on the level and activity of lactoferrin, lysozyme, antioxidant and fat in human milk. Those components remained stable regardless of time (6 weeks) and temperature (5 °C and 25 °C) of storage. The authors observed the decrease only for superoxidase dismutase activity, but it was a significantly lower decline than is observed in frozen raw human milk [24]. Results of our study and from the other authors emphasized the high potential of utilizing lyophilization for donor milk storage at human milk banks.

Despite these reports, a combination of human milk pascalization and lyophilization with subsequent storage is a potentially useful method that may be used in human milk banks. The combination of high-pressure processing and subsequent freeze-drying does not introduce significant changes in bioactivity and microbiological purity in human milk following long-term storage.

This is particularly desirable because of the growing need of hospitals for milk. The use of pressure and subsequent freeze-drying has a real impact on increasing milk resources in milk banks. Human milk, subjected to pascalization and subsequent lyophilization, can be easily stored in milk banks. Moreover, this method allows safe and fast transport of milk samples from milk banks over long distances without the fear of contamination or changes in the biological properties of this product. This is a particularly important aspect in the current pandemic situation caused by severe acute respiratory syndrome coronavirus 2 (SARS-CoV-2). The availability of human milk has drastically decreased, of course, due to the lack of donors but mainly due to the inability to distribute milk from stocks that are in banks [37,38,39]. Stored samples of human milk are preserved only by the HoP method, and therefore their shelf life is limited. The combination of HPP methods and lyophilization significantly extends the time of using stored milk samples, which gives a chance for the creation of local/global collection of samples and their use in situations of increased need.

## 5. Conclusions

HPP at 450 MPa is an effective method for treating human milk, as it preserves its bioactive components and microbiological purity. Furthermore, lyophilization is an effective method for storing human milk samples previously subjected to HPP at 450 MPa. This method of storing human milk is very practical and gives unlimited storage options for human milk. The application of our method unequivocally increases the availability of milk and satisfies, in a timely manner, the nutritional needs of all needy premature babies. Additionally, it significantly reduces the possibility of milk stored in milk banks being a source of pathogenic strains. The presented studies have significant methodological limitations resulting in a narrow spectrum of biological components having been analyzed and a lack of virus examination. More research is needed in line with the EMBA algorithm to obtain comparable results on the most promising techniques of human milk processing.

## Figures and Tables

**Figure 1 ijerph-18-02147-f001:**
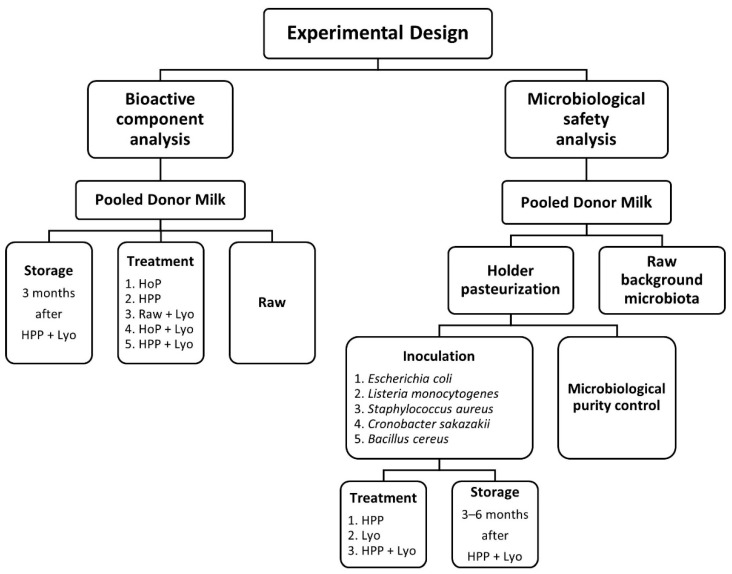
Experimental design including the analyses of bioactive components and microbiological safety in raw, treated, and stored human milk samples.

**Figure 2 ijerph-18-02147-f002:**
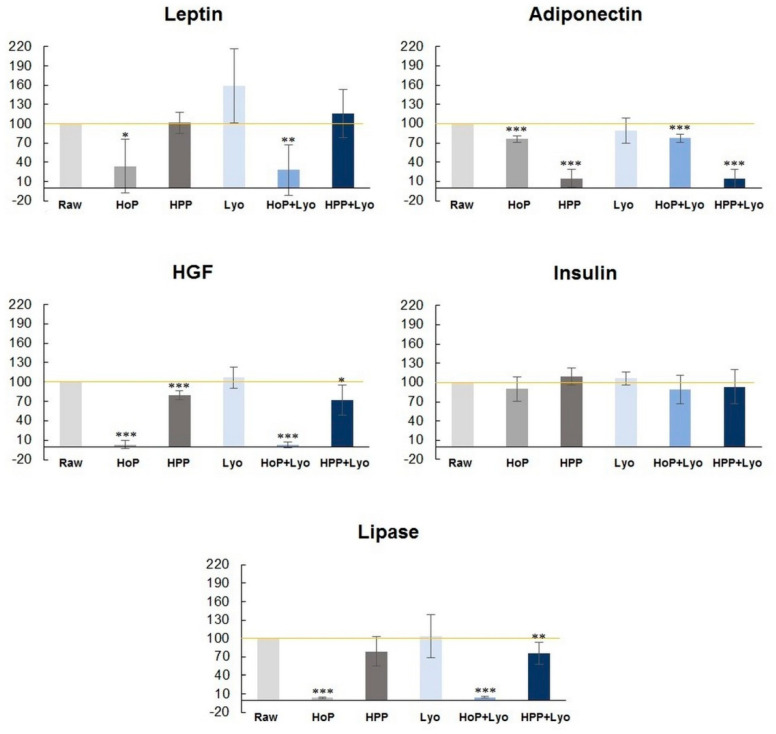
Changes in leptin, adiponectin, hepatocyte growth factor (HGF), insulin, and lipase concentrations (%) in processed human milk versus raw milk (Raw) (100%). The data are presented as mean values with the standard error of the mean. The percentage of the concentrations of the selected bioactive components in milk samples subjected to Holder pasteurization (HoP), high-pressure processing (HPP), freeze-drying (Lyo), HoP + Lyo, and HPP + Lyo (at time 0) compared to raw human milk (100%). Statistically significant differences are marked with asterisks, depending on the degree of significance: *** *p* < 0.0001, ** *p* < 0.001, * *p* < 0.05.

**Figure 3 ijerph-18-02147-f003:**
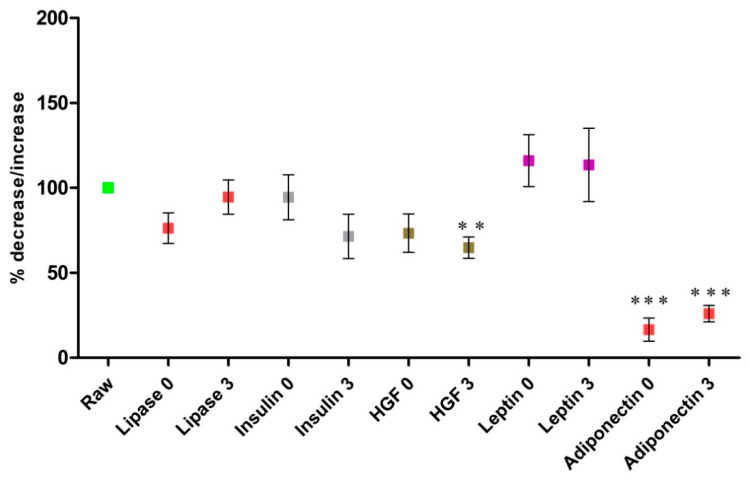
Changes in mean concentrations of leptin, adiponectin, hepatocyte growth factor (HGF), insulin, and lipase in human milk after the HPP + Lyo processes and 3-month storage compared with raw human milk. The data are presented as mean values with the standard error of the mean. The percentage of concentrations of the selected bioactive components in milk samples subjected to HPP + Lyo at time 0 (0) and at 3 months (3) compared to raw human milk (100%) (3). Statistically significant differences were observed and marked with asterisks, depending on the degree of significance (*** *p* < 0.0001, ** *p* < 0.001), only adiponectin (at time 0, *p* = 0.0002; at month 3, *p* < 0.0001) and HGF (at month 3, *p* = 0.0049). The differences in leptin, insulin, and lipase levels measured in treated (HPP + Lyo) milk samples stored for 3 months and in raw milk samples were not statistically significant (*p* > 0.05).

**Figure 4 ijerph-18-02147-f004:**
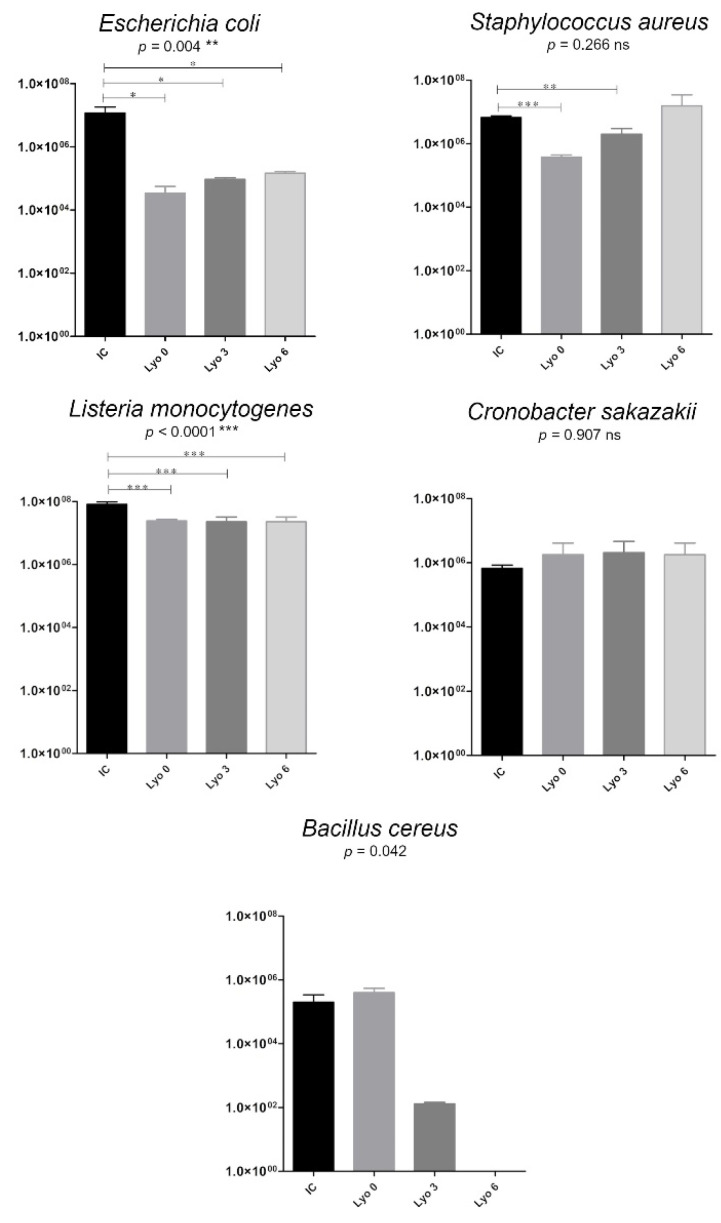
Changes in bacterial growth (colony-forming units (CFU)/mL) in human milk samples inoculated with bacteria strains, freeze-dried, and stored for up to 6 months. The data are presented as mean values with standard deviations. One-way analysis of variance (ANOVA) with post hoc Tukey’s test was used to statistically analyze the data from all replicates; Paired *t*-tests were used for the initial bacterial concentration (IC) and bacterial concentrations after freeze-drying at time 0 (Lyo 0), at 3 months of storage (Lyo 3), and at 6 months (Lyo 6) of storage. Statistically significant differences are marked with asterisks, depending on the degree of significance: *** *p* < 0.0001, ** *p* < 0.001, * *p* < 0.05, whereas non-significant differences (*p* > 0.05) are marked with ‘ns’.

**Table 1 ijerph-18-02147-t001:** Log bacterial growth in pooled human milk samples inoculated with bacterial strains, treated with pascalization or a combination of pascalization, and subjected to freeze-drying at time 0 and at 3 and 6 months of storage.

Genus	IC ^1^	HPP	HPP + Lyo 0	HPP + Lyo 3	HPP + Lyo 6	IE ^2^	%RG ^3^
*Staphylococcus aureus*	6.83 ± 0.03	0	0	0	0	6.83	100%
*Escherichia coli*	7.07 ± 0.13	0	0	0	0	7.07	100%
*Listeria monocytogenes*	7.91 ± 0.04	0	0	0	0	7.91	100%
*Bacillus cereus*	5.3 ± 0.08	0	0	0	0	5.3	100%
*Cronobacter sakazakii*	5.9 ± 0.24	0	0	0	0	5.9	100%

The data are presented as mean values with the standard error of the mean. ^1^ IC—initial concentration; ^2^ IE—inactivation efficiency (the decrease in bacterial growth by orders of magnitude after HPP and HPP + Lyo compared with IC); ^3^ %RG—the percentage reduction of bacterial growth compared with IC.

## Data Availability

Not applicable.

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
