# Peer review of "Combination of High-Pressure Processing and Freeze-Drying as the Most Effective Techniques in Maintaining Biological Values and Microbiological Safety of Donor Milk"

_ijerph, 2021, doi:10.3390/ijerph18042147_

Round 1

Reviewer 1 Report

Dr. Jarzyna and Collegues aimed at investigating novel methods for donor human milk processing.

This study addresses one crucial topi of preterm's nutrition and therefore is of great interest for the scientific community. However, there are some concerns that have to be addressed before it could be considered for publication.

OVERALL

The manuscript need an extensive english editing, since it's quite difficult to follow the research due to poor syntax and grammatical mistakes.

ABSTRACT

The abstract does not provide an adequate background, allowing the reader to understand the current policies and the novelty of the research. The abstract conclusions provide a statement only for High pressure treatment, while the title induces the reader in thinking that the combination of the two methods is the best alternative.

INTRODUCTION

The introduction does not describe adequately the current methods and their critical aspects. Therefore, is difficult to understand the comparison between the proposed alternative method. Moreover, it focuses on microbiological safety but does not offer description of the loss of immunological components, breastmilk cells (especially stem cells). I personally do not agree that "few previous studies on the safety of alternative human milk preservation methods yielded inconclusive results" as there is a growing body of literature on milk processing methods comparison and the 2 articles cited by the authors are not the most relevant.

METHODS and CONCLUSIONS

Since only bacteria strains have been investigated, without mentioning viruses, this article can not be conclusive on the safety of the alternative methods proposed. Moreover, the bioactive components investigated refer only to the hormonal and enzyme part of the functional substances of human milk. Therefore, it have to be specified in the title, abstract and overall that this manuscript refers only to a small section of milk components. 

DISCUSSION

I suggest the authors to reduce the lenght of the discussion, since a detailed description of the Holder pasteurization is misleading and should be briefly placed in the introduction

Author Response

Responses to the Reviewers’ Comments and Suggestions

Review of ijerph-1061679

Title: Combination of High-Pressure Processing and Freeze-Drying as the most Effective Techniques in Maintaining Biological Values and Microbiological Safety of Donor Milk

Authors: Sylwia Jarzynka , Kamila Strom, Olga Barbarska, Emilia Pawlikowska, Anna Minkiewicz-Zochniak, Elzbieta Rosiak, Gabriela Oledzka and Aleksandra Wesolowska

We are very grateful to the reviewers for their critical comments and thoughtful suggestions. Based on these comments and suggestions, we have made careful modifications to the original manuscript. All our textual changes are shown using a blue font based on the comments by all the reviewers. The point-to-point replies and explanations for all of the revisions are listed below for easy reference. Additionally, we attached the graphical abstract of our research. We hope that the revised manuscript can be published in the MDPI following these significant changes.

Major changes:

  • Altered the abstract to reflect the new structure of the manuscript.
  • Added several new references
  • Added a clearer description of strengths points and limitations of this study.
  • Provided more detail on the used technics.
  • Refreshed the text and ensuring a consistent structure it throughout the manuscript.
  • Added English Editing Certificate.

Reviewer #1

Dr. Jarzyna and Collegues aimed at investigating novel methods for donor human milk processing.

This study addresses one crucial topi of preterm's nutrition and therefore is of great interest for the scientific community. However, there are some concerns that have to be addressed before it could be considered for publication.

OVERALL

The manuscript need an extensive english editing, since it's quite difficult to follow the research due to poor syntax and grammatical mistakes.

Reply: We very much appreciate the reviewer’s suggestions, in the end, the paper has been edited for language by a company dedicated to helping international researchers publish their findings, by Wiley Editing Services (www.wileyeditingservices.com). We attached a certificate.

ABSTRACT

The abstract does not provide an adequate background, allowing the reader to understand the current policies and the novelty of the research. The abstract conclusions provide a statement only for High pressure treatment, while the title induces the reader in thinking that the combination of the two methods is the best alternative.

Reply and revision: We very much appreciate the reviewer’s detailed evaluations and suggestions. We have taken the comments on board to improve and clarify the manuscript. We have added further detail, and a better structure, to explain our choice of methods and make these explanations more easily understandable for those readers. Combination of both technics is discussed in the abstract of the revised manuscript.

The sentence now reads:

Abstract: Background: Human milk banks have a pivotal role in provide optimal food for those infants who are not fully breastfeed, by allowing to collect, processing and right distribute of human milk from donors. Donor human milk (DHM) is usually preserved by Holder pasteurization considered to be the gold standard to ensure microbiology safety and nutritional value of milk. However, as stated by the European Milk Banking Association (EMBA) there is a need to implement the improvement of the operating procedure of human milk banks including preserving and storing techniques. Aim: The purpose of this study was to assess the effectiveness and safety of the selected new combination of methods for preserving donor human milk in comparison with thermal treatment (Holder pasteurization). Methods: We assessed (1) the concentration of bioactive components (insulin, adiponectin, leptin, activity of pancreatic lipase, and hepatocyte growth factor) and (2) microbiological safety in raw and pasteurization, high-pressure processed and lyophilization human breast milk. Results: The combination of two techniques, high-pressure processing and freeze-drying, that showed the best potential for preserving the nutritional value of human milk and were evaluated for microbiological safety. Microbiological safety assessment excludes the possibility of using freeze-drying alone for human milk sample preservation. However, it can be used as a method for long-term storage of milk samples, which has previously been preserved via other processes. Conclusion: The results show that high-pressure treatment is the best method for preservation that ensures microbiological safety and biological activity but subsequent freeze -drying allowed for long term storage without loss of properties.   

INTRODUCTION

The introduction does not describe adequately the current methods and their critical aspects. Therefore, is difficult to understand the comparison between the proposed alternative method. Moreover, it focuses on microbiological safety but does not offer description of the loss of immunological components, breastmilk cells (especially stem cells). I personally do not agree that "few previous studies on the safety of alternative human milk preservation methods yielded inconclusive results" as there is a growing body of literature on milk processing methods comparison and the 2 articles cited by the authors are not the most relevant.

Reply and revision: We appreciate the reviewer’s comments. We agree with most of them, and the manuscript has been revised thoroughly according to the reviewer’s advice. We have also included several new references (see the end of message) which were either recommended for inclusion, by the reviewers or are necessary to include based on the textual changes recommended

Additional references:

Lozano, B.; Castellote, A.I.; Montes, R.; López-Sabater, M.C. Vitamins, fatty acids, and antioxidant capacity stability during storage of freeze-dried human milk. International Journal of Food Sciences and Nutrition 2014, 65, 703-707, doi:10.3109/09637486.2014.917154.

Oliveira, M.M.; Aragon, D.C.; Bomfim, V.S.; Trevilato, T.M.B.; Alves, L.G.; Heck, A.R.; Martinez, F.E.; Camelo, J.S., Jr. Development of a human milk concentrate with human milk lyophilizate for feeding very low birth weight preterm infants: A preclinical experimental study. PLoS One 2019, 14, e0210999, doi:10.1371/journal.pone.0210999.

Martysiak-Żurowska, D.; Rożek, P.; Puta, M. The effect of freeze-drying and storage on lysozyme activity, lactoferrin content, superoxide dismutase activity, total antioxidant capacity and fatty acid profile of freeze-dried human milk. Drying Technology 2020, 10.1080/07373937.2020.1824188, 1-11, doi:10.1080/07373937.2020.1824188.

METHODS and CONCLUSIONS

Since only bacteria strains have been investigated, without mentioning viruses, this article can not be conclusive on the safety of the alternative methods proposed. Moreover, the bioactive components investigated refer only to the hormonal and enzyme part of the functional substances of human milk. Therefore, it have to be specified in the title, abstract and overall that this manuscript refers only to a small section of milk components. 

Reply and revision: We thank the reviewer for these comments. We discuss the limitation of the study in the section discussion. We mansion, in this study bioactivity, was examined only selective, recognizing Lipase as one of the most important, which is completely deactivated by HP. We also admitted that not all infectious agents were examined in our study. Of course,  viruses can also be transmitted through breast milk, eg Human immunodeficiency virus (HIV), cytomegalovirus (CMV), Human T cell leukaemia virus (HTLV). These infections can be passed on from mothers with high viral load, and in milk banks, all donors are tested for major viruses. EMBA guides recommend testing of viruses in human milk to evaluate the new technology for processing human milk. However, we have no laboratory condition to conduct such research and these infectious agents are not covered by our research. Regarding the of the results on specific technics, we would like to emphasize that the present study is the first analyze of human milk microbiology in the line with the methodology of EMBA recommendations ( Moro, G.E.; Billeaud, C.; Rachel, B.; Calvo, J.; Cavallarin, L.; Christen, L.; Escuder-Vieco, D.; Gaya, A.; Lembo, D.; Wesolowska, A., et al. Processing of Donor Human Milk: Update and Recommendations From the European Milk Bank Association (EMBA). Front Pediatr 2019, 7, 49, doi:10.3389/fped.2019.00049). These limitations and need of further research have been stressed in conclusion section (line 477:487).

DISCUSSION

I suggest the authors to reduce the lenght of the discussion, since a detailed description of the Holder pasteurization is misleading and should be briefly placed in the introduction

Reply and revision: We agree with the reviewer here and have revised the manuscript especially the section's discussion. These parts have now been modified and rephrased in the introduction session accordingly.

Reviewer 2 Report

The authors are exploring methods to determine the best practices to ensure not only good quality milk, but also milk low in microbial contamination.

The paper was difficult to follow in some sections and may not be as connected. The authors begin by exploring the bioactivity of milk, examining how various treatments impacted the levels of specific bioactive molecules. The authors then review microbial safety of the milk. They indicate that milk was assayed for existing microbes, but this data was not present. This is important as it provides a baseline. It was unclear why the authors felt the need to inoculate the milk.

Although a flow chart was provided, there was no real rationale for the experiments selected, which made following the outcome difficult.

Author Response

Responses to the Reviewers’ Comments and Suggestions

Review of ijerph-1061679

Title: Combination of High-Pressure Processing and Freeze-Drying as the most Effective Techniques in Maintaining Biological Values and Microbiological Safety of Donor Milk

Authors: Sylwia Jarzynka , Kamila Strom, Olga Barbarska, Emilia Pawlikowska, Anna Minkiewicz-Zochniak, Elzbieta Rosiak, Gabriela Oledzka and Aleksandra Wesolowska

We are very grateful to the reviewers for their critical comments and thoughtful suggestions. Based on these comments and suggestions, we have made careful modifications to the original manuscript. All our textual changes are shown using a blue font based on the comments by all the reviewers. The point-to-point replies and explanations for all of the revisions are listed below for easy reference. Additionally, we attached the graphical abstract of our research. We hope that the revised manuscript can be published in the MDPI following these significant changes.

Major changes:

  • Altered the abstract to reflect the new structure of the manuscript.
  • Added several new references
  • Added a clearer description of strengths points and limitations of this study.
  • Provided more detail on the used technics.
  • Refreshed the text and ensuring a consistent structure it throughout the manuscript.
  • Added English Editing Certificate.

Reviewer #2

Comments and Suggestions for Authors

The authors are exploring methods to determine the best practices to ensure not only good quality milk, but also milk low in microbial contamination.

The paper was difficult to follow in some sections and may not be as connected. The authors begin by exploring the bioactivity of milk, examining how various treatments impacted the levels of specific bioactive molecules. The authors then review microbial safety of the milk. They indicate that milk was assayed for existing microbes, but this data was not present. This is important as it provides a baseline. It was unclear why the authors felt the need to inoculate the milk.

Although a flow chart was provided, there was no real rationale for the experiments selected, which made following the outcome difficult.

Reply and revision: We thank reviver for their careful reading of our work and thoughtful comments. The manuscript has now been modified and rephrased. We believe our substantial restructuring has helped make the aims, methods and results more transparent. We agree that the manuscript would benefit from a brief discussion of selected experiments. We direct the reader to other papers such as Lozano, B.; 2014, and Oliveira, M.M.; 2019, and Martysiak-Żurowska, D.; 2020. We have also ensured that we explain the limitation of the study in the section introduction (line 121:128).

We add also additional information about microbiological tests. The sentence now reads:

“Microbiological tests were performed for microorganisms that could cause gastrointestinal, respiratory tract, and systemic infections, including meningitis and sepsis, in neonates and infant. Due to the increase in contamination of milk with potentially pathogenic bacteria, in our microbiological tests, we used the method of milk fortification with the selected pathogen and then treating these samples with various techniques that can be used in the safe preparation and storage of human milk in milk banks. The goal was to find the safest method in terms of preserving important hormones and metabolic enzymes, and the most effective in terms of microbiological purity.”

Reviewer 3 Report

This paper describes a study of the impact of physical processes (high pressure, heat, and freeze-drying) on the stability of human milk.  Preservation of human milk in milk banks and for storage and distribution is a major topic of interest to paediatric and neonatal care internationally, and the idea that new processing solutions exist for making such a valuable biological material safe and stable is topical and of broad interest. 

The study presented appears to have been well conducted using appropriate methods, and the paper is well written, so I am happy to recommend it for publication following consideration of only a few minor points:

  1. Handling of the freeze-dried samples is not very clearly presented. For example, what analysis was done in dry or reconstituted form, and how was reconstitution done?  Was the 3 month storage in dry form or were some liquid samples stored this long too (as this is very long obviously if not dried)?  This area just needs better explanation
  2. It would be useful to explain why the selected bioactive compounds were measured, as the significance of leptin and adiponectin, in particular, and why they need to be preserved, is not very explicitly explained

Author Response

Responses to the Reviewers’ Comments and Suggestions

Review of ijerph-1061679

Title: Combination of High-Pressure Processing and Freeze-Drying as the most Effective Techniques in Maintaining Biological Values and Microbiological Safety of Donor Milk

Authors: Sylwia Jarzynka , Kamila Strom, Olga Barbarska, Emilia Pawlikowska, Anna Minkiewicz-Zochniak, Elzbieta Rosiak, Gabriela Oledzka and Aleksandra Wesolowska

We are very grateful to the reviewers for their critical comments and thoughtful suggestions. Based on these comments and suggestions, we have made careful modifications to the original manuscript. All our textual changes are shown using a blue font based on the comments by all the reviewers. The point-to-point replies and explanations for all of the revisions are listed below for easy reference. Additionally, we attached the graphical abstract of our research. We hope that the revised manuscript can be published in the MDPI following these significant changes.

Major changes:

  • Altered the abstract to reflect the new structure of the manuscript.
  • Added several new references
  • Added a clearer description of strengths points and limitations of this study.
  • Provided more detail on the used technics.
  • Refreshed the text and ensuring a consistent structure it throughout the manuscript.
  • Added English Editing Certificate.

Reviewer #3

Comments and Suggestions for Authors

This paper describes a study of the impact of physical processes (high pressure, heat, and freeze-drying) on the stability of human milk.  Preservation of human milk in milk banks and for storage and distribution is a major topic of interest to paediatric and neonatal care internationally, and the idea that new processing solutions exist for making such a valuable biological material safe and stable is topical and of broad interest. 

The study presented appears to have been well conducted using appropriate methods, and the paper is well written, so I am happy to recommend it for publication following consideration of only a few minor points:

  1. Handling of the freeze-dried samples is not very clearly presented. For example, what analysis was done in dry or reconstituted form, and how was reconstitution done?  Was the 3 month storage in dry form or were some liquid samples stored this long too (as this is very long obviously if not dried)?  This area just needs better explanation
  2. It would be useful to explain why the selected bioactive compounds were measured, as the significance of leptin and adiponectin, in particular, and why they need to be preserved, is not very explicitly explained

Reply and revision: We would like to thank the reviewer for their comments and suggestions for the manuscript. We believe that the comments have identified important areas which required improvement. After completion of the suggested edits, the revised manuscript has benefitted from an improvement in the overall presentation and clarity.

We agree that more precise technical formulations clarify the description, so we added additional information to the text regarding used methods. We also clarified information about the parameters of storing milk samples. Samples had been processing in five replicates, stored for 3 months and 6 months frozen in -20 °C or the freeze-drying form at refrigerator temperature. Microbiological tests were carried out following the European standards for testing food products on thawed samples or hydrated samples after freeze-drying before processing. Regarding the of the results on specific technics, we would like to emphasize that the present study is the first analyze of human milk to be in the line with the methodology of EMBA recommendations ( Moro, G.E.; Billeaud, C.; Rachel, B.; Calvo, J.; Cavallarin, L.; Christen, L.; Escuder-Vieco, D.; Gaya, A.; Lembo, D.; Wesolowska, A., et al. Processing of Donor Human Milk: Update and Recommendations From the European Milk Bank Association (EMBA). Front Pediatr 2019, 7, 49, doi:10.3389/fped.2019.00049.)  We choose lipase, leptin, adiponectin and insulin because these are the most important factors in energy metabolisms, food intake and appetite regulation, which constitute a leading role in newborns nutrition. Lipase and adiponectin additionally stimulate the immune system of neonates and have also cardioprotective effects and intake role in bone formation. HGF factor is involved in regulating the growth of intestinal cells in the newborn.
